# Thermotolerance in the pathogen *Cryptococcus neoformans* is linked to antigen masking via mRNA decay-dependent reprogramming

Amanda L.M. Bloom[1], Richard M. Jin[1], Jay Leipheimer[1], Jonathan E. Bard [2], Donald Yergeau[2], Elizabeth A. Wohlfert[1] & John C. Panepinto[1]*

A common feature shared by systemic fungal pathogens of environmental origin, such as *Cryptococcus neoformans*, is their ability to adapt to mammalian core body temperature. In *C. neoformans*, this adaptation is accompanied by Ccr4-mediated decay of ribosomal protein mRNAs. Here we use the related, but thermo-intolerant species *Cryptococcus amylolentus* to demonstrate that this response contributes to host-temperature adaptation and pathogenicity of cryptococci. In a *C. neoformans ccr4Δ* mutant, stabilized ribosomal protein mRNAs are retained in the translating pool, and stress-induced transcriptomic changes are reduced in comparison with the wild type strain, likely due to ineffective translation of transcription factors. In addition, the mutant displays increased exposure of cell wall glucans, and recognition by Dectin-1 results in increased phagocytosis by lung macrophages, linking mRNA decay to adaptation and immune evasion.

[1] Department of Microbiology and Immunology, Witebsky Center for Microbial Pathogenesis and Immunology, Jacobs School of Medicine and Biomedical Sciences, University at Buffalo, SUNY, Buffalo, NY 14203, USA. [2] New York State Center of Excellence and Bioinformatics, University at Buffalo, SUNY, Buffalo, NY 14203, USA. *email: jcp25@buffalo.edu

The interaction between a host and an infecting micro-organism results in disease when damage inflicted by the microorganism or the immune response reaches an observable level[1]. Compromised immune function creates opportunities for an invading organism to establish deep infection with the requirement that the infectious agent is equipped to adapt to the host environment and defend against any intact innate immune function. Of the myriad of fungi in the environment, few are able to cause systemic infections in humans. For those that do, a human host is not required for survival, and pathogenicity is suggested to be accidental[2]. A major restriction to fungi establishing human infections is mammalian core body temperature, and a unifying feature of systemic fungal pathogens of environmental origin is thermotolerance.

*Cryptococcus neoformans* is an encapsulated environmental fungus and accidental pathogen[2] responsible for over 200,000 worldwide deaths annually[3]. *C. neoformans* is regularly inhaled, but in healthy individuals is rapidly cleared and/or maintained in a dormant state[4]. In the instance of compromised immunity, however, *C. neoformans* disseminates from the lung to the central nervous system causing deadly meningoencephalitis. Interactions with resident alveolar macrophages play a key role in determining the fate of the pathogen. Classical activation of macrophages promotes Th1-skewed proinflammatory responses and clearance of the pathogen, whereas alternatively activated macrophages lead to Th2 responses that promote intracellular survival[5]. In other fungi, Dectin-1 recognition of cell wall β-(1,3)-glucan elicits Th17 responses as well[6–9]. Unlike these fungi, Th17 signaling plays a minimal role in host defense to *C. neoformans*[10], likely because its beta-glucans are masked by other cell wall components and capsule.

Environmental fungal pathogens have likely acquired the requisite adaptive mechanisms through interactions in their environmental niche. For example, the ability of *C. neoformans* to thrive within a macrophage phagosome has been attributed to interactions with predatory amoeba in the soil[11]. It follows, then, that increasing global surface temperatures may select for thermotolerance, and concomitantly may lead to the emergence of new fungal pathogens[12]. Understanding molecular programs involved in temperature adaptation will be important for combating infections caused by current pathogens as well as new pathogens that could emerge as a result of environmental selection. *Cryptococcus amylolentus* is an environmental fungus and the closest known relative of the pathogenic Cryptococci[13]. *C amylolentus* was previously reported to have "virulence potential", possessing the ability to produce capsule and demonstrating infectivity and pathogenicity in invertebrate hosts at environmental temperature. However, *C. amylolentus* is avirulent in invertebrate hosts at 37 °C and in mice due to thermointolerance[14]. Given this, *C. amylolentus* is a useful tool to define the molecular processes necessary for host-temperature adaptation in the Cryptococci.

In response to host temperature, mRNAs encoding ribosomal proteins (RPs) are rapidly degraded in *C. neoformans*[15,16]. This enhanced mRNA decay lasts 1 h, contributing to a transient repression of RP transcripts. In a mutant strain lacking the major mRNA deadenylase, Ccr4, mRNA decay is defective and RP transcripts are stabilized with steady-state levels remaining elevated. In addition, when grown at host temperature, the *ccr4Δ* mutant displays a budding defect and is attenuated in growth and virulence[17]. The specific contribution of RP transcript degradation and mRNA decay in host-temperature adaptation is not fully understood. Studies in model yeasts have reported that RP transcript repression is part of a general environmental stress response (ESR) that occurs following many types of stress[18,19], and repression of these abundant mRNAs is suggested to reduce

competition for translational machinery and promote protein production of stress-induced transcripts[20].

Here, we report that immediate RP transcript decay and repression is absent in *C. amylolentus* following host-temperature stress, suggesting that this response may be necessary for host-temperature adaptation and pathogenicity. We set out to determine the fate of erroneously stabilized transcripts and the downstream consequences of misregulated mRNA decay in *C. neoformans* to better understand the importance and contribution of this mechanism of gene expression control. In the absence of *CCR4*, stabilized RP transcripts remain associated with the translational machinery, and several functional classes of genes are repressed following host-temperature stress. Transcripts encoding cell wall remodeling proteins and transcriptional regulators are among those repressed in the *ccr4Δ* mutant. We show that following host-temperature stress, cell wall glucans are exposed and recognized by macrophages and Dectin-1 in a *ccr4Δ* mutant. Among the transcriptional regulators repressed in the *ccr4Δ* mutant, one, *HOB1*, was found to promote antigen masking. The work presented here suggests that regulated mRNA decay is a pathogen-specific response to host-temperature stress in cryptococci that promotes adaptation via regulation of transcriptome and translatome reprogramming, and demonstrates a link between temperature adaptation and innate immune evasion in the Cryptococci.

## Results

**RP mRNA decay correlates with host-temperature adaptation.** To determine if temperature-induced rapid degradation of RP transcripts was specific to *C. neoformans*, we assessed RP transcript decay in the thermointolerant relative *C. amylolentus*. *RPL2* levels and decay rates do not change significantly following a shift to 37 °C in *C. amylolentus* (least squares fit of exponential decay nonlinear regression, $F_{1,28} = 0.2798$, $p = 0.601$), in contrast to *C. neoformans*, in which *RPL2* decays rapidly ($F_{1,28} = 30.57$, $p < 0.0001$; Fig. 1a[15]).

In response to carbon starvation, RP transcripts are destabilized in wild-type *C. neoformans*, but remain stable in the *ccr4Δ* mutant[21]. To determine if the absence of a decay response in *C. amylolentus* is specific to temperature, or is absent in response to stress in general, we monitored *RPL2* levels following carbon starvation in *C. amylolentus*. RP transcripts diminished upon a shift to YNB minimal media lacking dextrose in *C. amylolentus*, albeit at a slightly slower rate than in *C. neoformans* (Supplementary Fig. 1a[21]). Furthermore, RP transcript half-lives decrease more rapidly during carbon starvation in comparison with YPD in *C. amylolentus* (least squares fit of exponential decay nonlinear regression, $F_{1,28} = 138.6$, $p < 0.0001$), suggesting that the decay response to starvation is conserved between the species (Supplementary Fig. 1b).

The difference in thermotolerance between *C. neoformans* and *C. amylolentus* correlates to a lack of mRNA decay in response to host-temperature stress. This difference could arise from either the mRNA decay machinery, itself, or from differences in upstream signals. To address this question, we expressed the *CCR4* gene from *C. amylolentus* with the promoter, 5′UTR, and terminator region of the *C. neoformans CCR4* gene with a neomycin resistance cassette in the *C. neoformans ccr4Δ* mutant (Supplementary Fig. 2). BLAST analysis demonstrated that the Ccr4 homologs share 79% protein identity and contain the conserved leucine-rich repeat, required for protein binding, and the catalytic EEP domain (Supplementary Fig. 2). Expression of the *C. amylolentus* Ccr4 in *C. neoformans ccr4Δ* complemented the temperature sensitivity on agar plates (Fig. 2a) and restored growth in liquid media at 37 °C, although growth rate

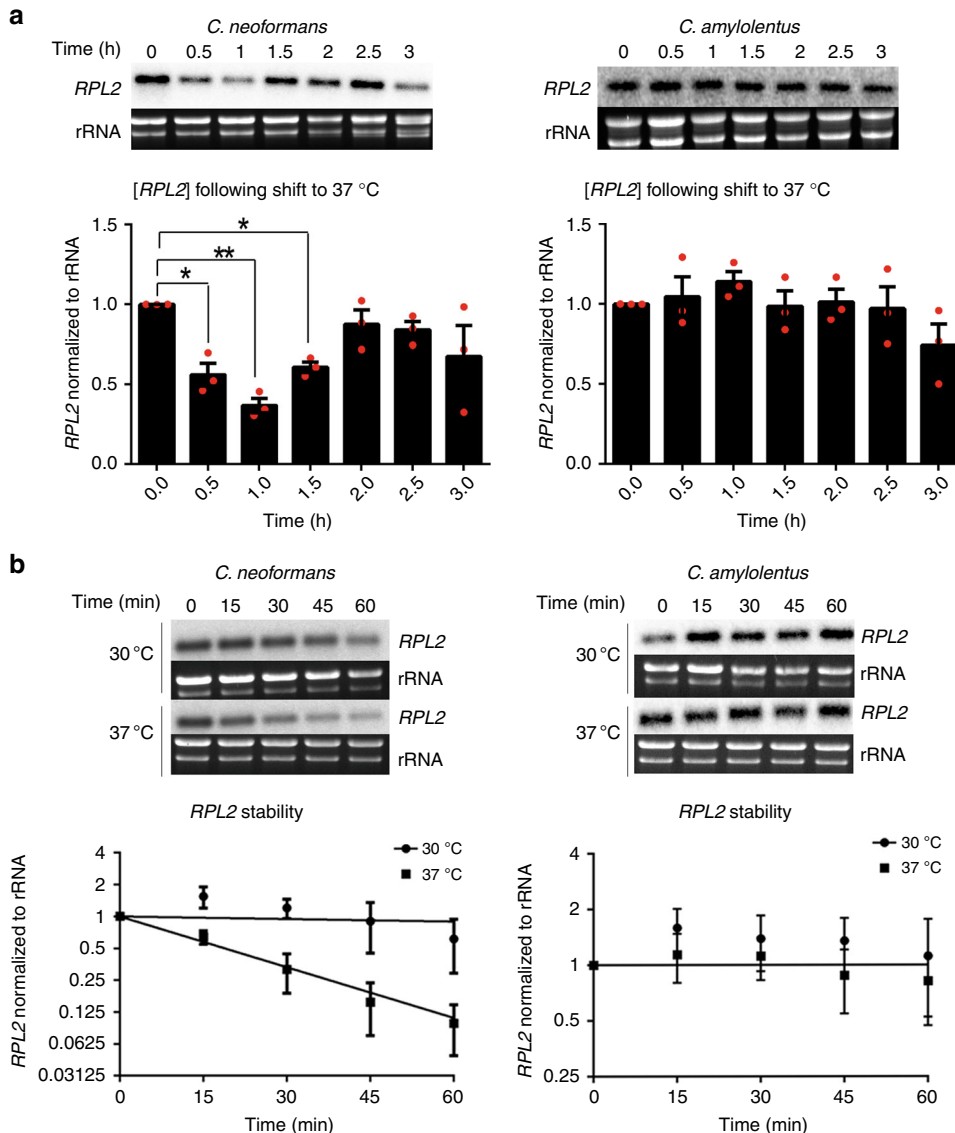

**Fig. 1** Enhanced RP transcript decay and repression is absent in *C. amylolentus*. **a** Wild-type *C. neoformans* and *C. amylolentus* strains were grown to mid-log phase in YPD at 30 °C and then shifted to pre-warmed 37 °C YPD. RNA was extracted from samples every 30 min for 3 h for northern blot assessment of *RPL2* levels. Graphs show mean *RPL2* levels ± s.e.m. normalized to rRNA, $n = 3$. Expression of *RPL2* at each time point was compared with $t = 0$ for each strain by ANOVA with Dunnett's test post hoc. **b** Stability of *RPL2* at 30 °C and following a shift to host temperature was assessed by northern blot analyses in wild-type *C. neoformans* and *C. amylolentus*. Transcription was inhibited by the addition of 1,10-phenanthroline, and RNA was extracted from samples collected every 15 min for 60 min. Graphs show the log2 regression of *RPL2* normalized to rRNA at 30 °C and 37 °C, $n = 3$. Significance between curves was detected by sum-of-squares F test, with $p < 0.05$ determining significant difference between regression lines. Source data are provided as a Source Data file

was observably slower than WT and the *C. neoformans CCR4* complement (Fig. 2b). The *C. amylolentus CCR4* complemented strain restored the transient repression of overall *RPL2* levels in response to host temperature (Fig. 2c), and exhibited enhanced decay of RP transcripts after a shift to 37 °C in comparison with the *ccr4Δ* mutant (least squares fit of exponential decay nonlinear regression, $F_{1,28} = 14.17$, $p = 0.0008$; Fig. 2d), but at a rate slower than that of wild-type *C. neoformans* or the *C. neoformans CCR4* complement. This suggests that the *C. amylolentus* Ccr4 is capable of initiating mRNA decay in *C. neoformans*, but that it may not function as efficiently as the *C. neoformans* protein. These results also suggest that there are signals upstream of Ccr4 in *C. neoformans* that activate mRNA decay in response to host temperature that are absent in *C. amylolentus*.

**Ccr4-mediated mRNA decay balances cellular mRNA levels**. Temperature stress leads to introduction of stress response mRNAs into the transcriptome through the activation of transcription factors. For example, the endoplasmic reticulum (ER) stress response is immediately induced following a shift to host temperature[22]. Because RP transcripts are not rapidly repressed upon host-temperature stress in the *ccr4Δ* mutant, but induction of the ER stress response is intact, we hypothesized that host-temperature stress could create an imbalance of mRNA in the cell. To quantify cellular mRNA, equivalent amounts of the total RNA were hybridized with a labeled oligo-dT probe. Levels of mRNA in the wild-type at 30 °C and 1 h after a shift to 37 °C were equivalent (Fig. 3a), suggesting that repression of some genes and induction of others is regulated to balance the total mRNA levels. While the total mRNA is equivalent between wild-type and the

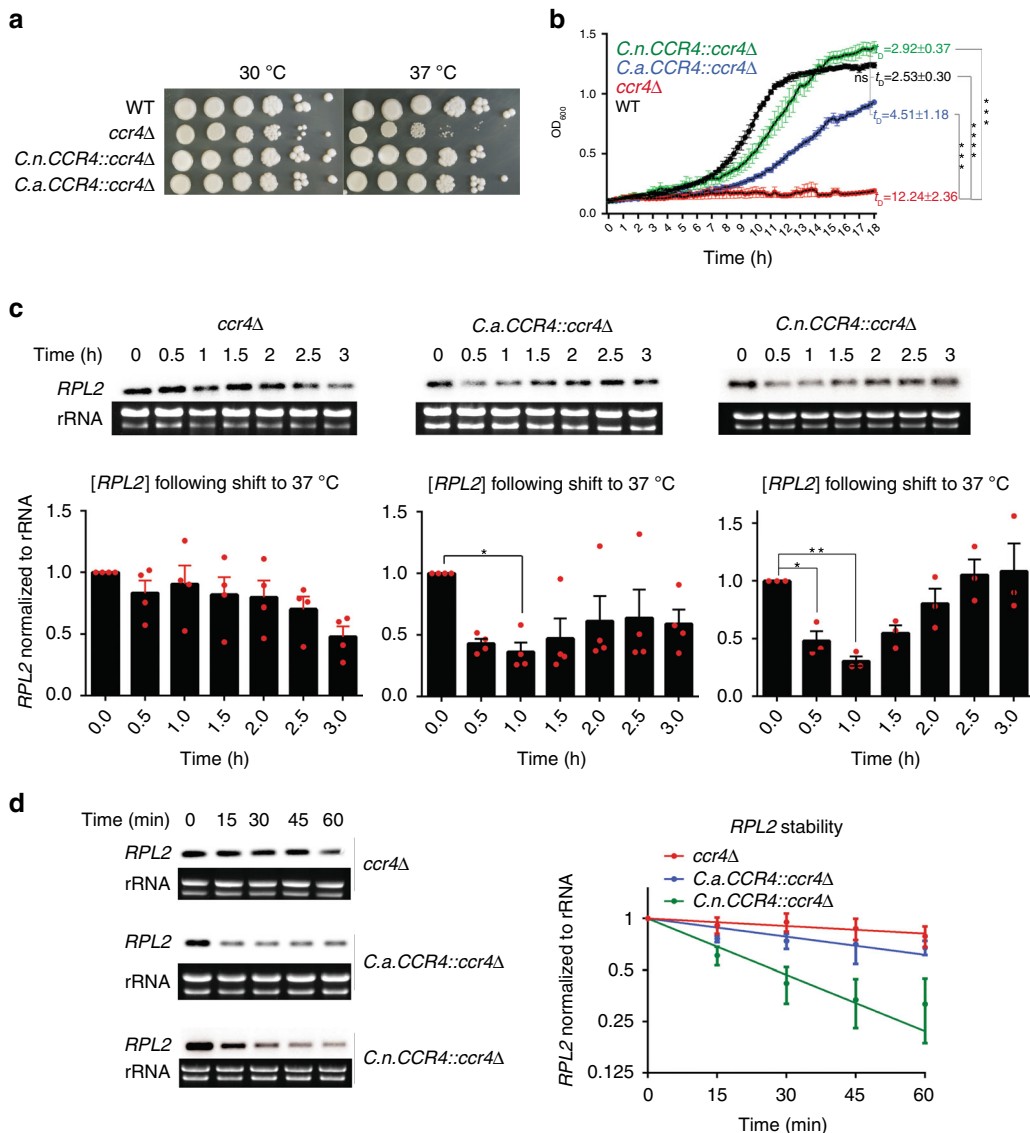

**Fig. 2** *C. amylolentus* Ccr4 is functional and responds to 37 °C stress. **a** Cultures of wild-type *C. neoformans*, the *C. neoformans ccr4Δ* mutant, the *C. neoformans (C.n.) CCR4::ccr4Δ*, and the *ccr4Δ* mutant strain expressing the *C. amylolentus CCR4* (*C.a.CCR4::ccr4Δ*) gene were adjusted to $OD_{600} = 1.0$ followed by tenfold serial dilution. In total, 5 μl of each dilution was spotted onto YPD agar plates and incubated at either 30 °C or 37 °C for 3 days. Images are representative of three biological replicates. **b** Growth curve analysis. Strains were incubated at 37 °C with orbital shaking for 18 h. Plotted are $OD_{600} \pm$ s.e.m. from 15 min increments; $n = 3$. Doubling times ($t_D$) during exponential growth were compared by ANOVA and Tukey's test post hoc, $n = 3$. **c** *RPL2* expression following a shift to host temperature for 3 h was assessed by northern blot analyses. Graphs show mean *RPL2* levels ± s.e.m. normalized to rRNA, $n \geq 3$. Expression of *RPL2* at each time point was compared with $t = 0$ for each strain by ANOVA with Dunnett's test post hoc. **d** *RPL2* stability following a shift to 37 °C in the presence of the transcriptional inhibitor, 1,10-phenanthroline, was assessed by northern blot analyses. Graphs show the log2 regression of *RPL2* normalized to rRNA, $n = 3$. Significance between curves was detected by sum-of-squares F test, with $p < 0.05$ determining significant difference between regression lines. Source data are provided as a Source Data file

*ccr4Δ* mutant in the absence of stress, there was a significant increase following stress in the *ccr4Δ* mutant, suggesting that mRNA decay is needed to balance the level of cellular mRNA during stress.

**Stabilized RP transcripts fail to leave the translating pool**. The fate of stabilized RP transcripts in the *ccr4Δ* mutant and their impact on the response to temperature stress is unknown. There is evidence that deadenylation occurs on the ribosome and results in repression of mRNA translation[23]. To determine if stabilized transcripts remain associated with actively translating ribosomes, we examined polysome profiles of cells grown at 30 °C or shifted

to 37 °C for 1 h and assessed *RPL2* distribution across the polysome gradient. Northern blot analyses of collected fractions reflected reduction of overall levels of *RPL2* in the wild-type following stress and revealed retention of *RPL2* in the polysomes of the *ccr4Δ* mutant following the shift to 37 °C (Fig. 3b). Interestingly, we also observed some translational repression in the wild-type following stress, but the translational profile in the *ccr4Δ* mutant was unchanged.

We were curious how abundant, stabilized RP transcripts retained in the polysome fractions might impact the entry of stress-induced transcripts into the translating pool. We hypothesized that RP transcripts may outcompete newly synthesized stress-responsive transcripts for the translational machinery

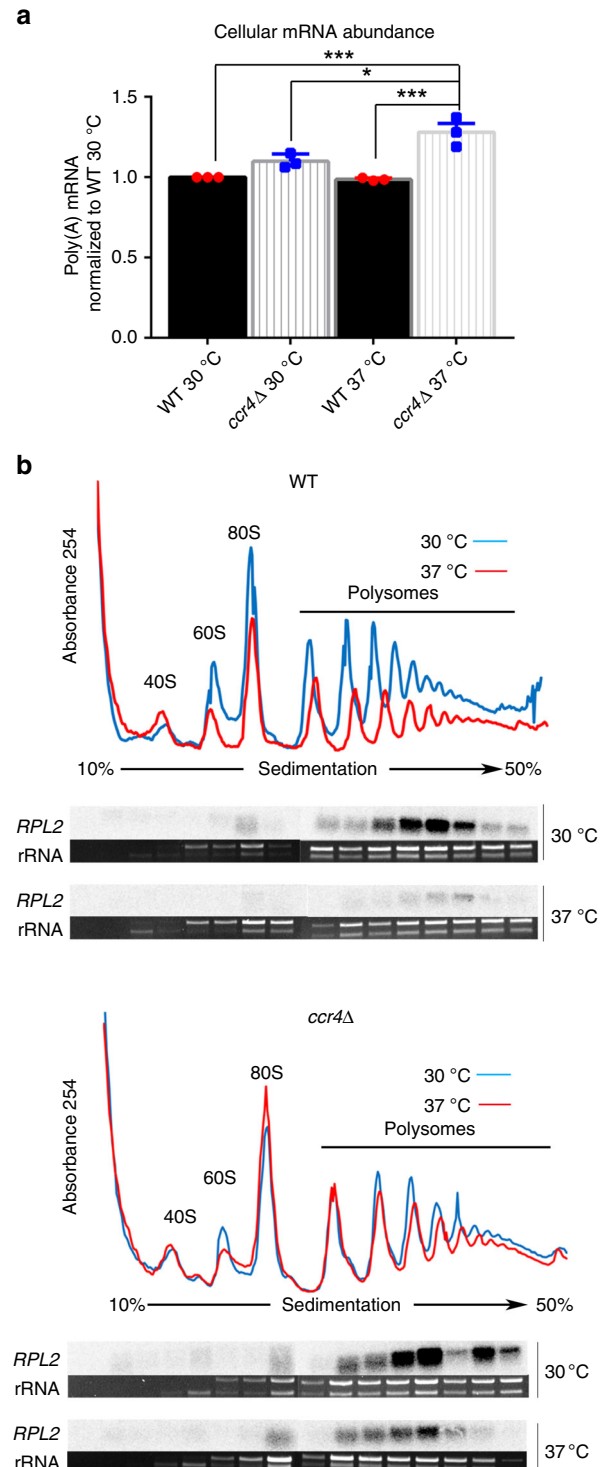

**a**

Cellular mRNA abundance

**b**

WT

*ccr4Δ*

**Fig. 3** Stabilized RP transcripts are retained in the actively translating pool. **a** Equivalent amounts of the total RNA from cultures grown at 30 °C or shifted to 37 °C for 1 h were vacuum filtered through a slot blot apparatus onto the nylon membrane. A P32-labeled oligo-dT was used to assess mRNA by northern blot. Graphs show averaged polyA mRNA levels ± s.e.m. from three biological replicates. **b** Polysome traces show the translational state of each strain during no stress and following a shift to 37 °C for 1 h. Subunits, monosomes, and polysome are indicated. RNA was extracted from fractions collected during polysome profiling for wild-type and *ccr4Δ*. Equivalent volumes of RNA from each fraction were run on a formaldehyde-agarose gel for northern blot analysis of *RPL2*. The rRNA bands confirm that the fractions correspond to the area of the polysome profile above. The results shown are representative of three biological replicates. Source data are provided as a Source Data file

of our comparisons are summarized in Table 1. Changes seen in the total mRNA (input) were mostly reflective of the changes seen in the translating pool (polysome) in both wild-type and *ccr4Δ*. Furthermore, transcripts that increased in abundance in the input were also increased in the polysome (Table 1, Fig. 4a), suggesting that entry of newly synthesized transcripts into the translating pool is not inhibited by RP transcript retention.

Consistent with our previous reports, our RNA-seq data revealed that RP transcripts were significantly downregulated in the wild-type, but not in the *ccr4Δ* mutant following host-temperature stress (Supplementary Dataset 1). Furthermore, our analyses also showed that stabilized RP transcripts in the *ccr4Δ* mutant are retained in the translating pool, reinforcing our previous *RPL2* data. Figure 4b is a graphical representation of the genes that are significantly changed in the translatome from 30 °C to 37 °C in the wild-type, plotted from greatest positive change to greatest negative change in magnitude (blue circles). Overlaid are the changes for each of these genes in the translatome in the *ccr4Δ* mutant (orange circles) with the RP transcripts highlighted (green) demonstrating that while RP transcripts leave the translating pool in the wild-type, they are retained and unchanged in the *ccr4Δ* mutant.

**mRNA decay is required for transcriptional reprogramming.** Our RNA-seq data revealed drastically different transcriptomes and corresponding translatomes between the WT and *ccr4Δ* mutant after the shift to host temperature (Table 1; Supplementary Table 1). Following stress ~3500 genes were differentially regulated (Table 1). Examining the changes that occurred from 30 °C to 37 °C within each strain individually further revealed misregulation in the absence of *CCR4*. While ~1400 genes undergo significant change following host-temperature stress in the wild-type, ~2500 are significantly changed in the *ccr4Δ* mutant, and only 452 genes undergo change in both strains (Table 1, Fig. 4c). The heatmap in Fig. 4a shows normalized expression of the 250 most variable genes in our data set. Several genes were downregulated in the WT following stress, but remained unchanged in the mutant, likely due to defective destabilization (Sections III, V, and VI). Unexpectedly, we observed transcriptional induction of several genes in the mutant that were unchanged in the WT (section I), suggesting that different transcriptional programs are activated either due to the absence of Ccr4 or in response to downstream consequences of defective mRNA decay. In addition, we observed genes that were upregulated in both strains, but were much higher in the mutant (section IV). Among these genes are those encoding heat-shock proteins and chaperones, and genes involved in trehalose metabolism (Supplementary Dataset 1). It is plausible that these genes, like the ER stress genes, may require deadenylation-dependent

resulting in their exclusion from the translating pool. We set out to examine how the "translatome", or pool of polysome-associated mRNAs, is reprogrammed during temperature adaptation. RNA was isolated from total input and polysome fractions from wild-type and the *ccr4Δ* mutant during no stress and following host-temperature stress for 1 h in triplicate. Input and polysome libraries were sequenced and the change in expression for each genomic locus at 30 °C and 37 °C, and between the two strains, was calculated. Principal component analysis demonstrated that three biological replicates for each sample were tightly clustered (Supplementary Fig. 3). Significant differences for each

| Table 1 Summarized results from RNA-seq analysis | | | | | | |
|---|---|---|---|---|---|---|
| Comparison | DEGs input | Down[a] in input | Up[a] in input | DEGs polysome | Down[a] in polysome | Up[a] in polysome |
| WT 30 °C vs. WT 37 °C | 1399 | 721 | 678 | 1390 | 568 | 822 |
| ccr4Δ 30 °C vs. ccr4Δ 37 °C | 2518 | 1198 | 1320 | 2726 | 1250 | 1476 |
| WT 30 °C vs. ccr4Δ 30 °C | 407 | 223 | 184 | 395 | 200 | 195 |
| WT 37 °C vs. ccr4Δ 37 °C | 3521 | 1672 | 1849 | 3599 | 1841 | 1758 |

| Comparison | DEGs |
|---|---|
| WT input vs. WT polysome 30 °C | 138 |
| WT input vs. WT polysome 37 °C | 2 |
| ccr4Δ input vs. ccr4Δ polysome 30 °C | 360 |
| ccr4Δ input vs. ccr4Δ polysome 37 °C | 195 |

[a]Up/down refers to expression change after shift to 37 °C; up/down refers to in WT in instances when strains are compared

decay to dampen their induced levels. Our data also revealed genes that were higher in the mutant under no stress, and expression increased more in the mutant after stress (section II). The vast amount of upregulation seen in this data set is in the ccr4Δ mutant, demonstrating further that mRNA decay is needed for balancing cellular mRNA abundance.

Though not reflected in the 250 most variable genes, there was a large amount of DEGs with lower expression in the ccr4Δ mutant compared with WT at 37 °C (Supplementary Fig. 4). This was surprising, but further suggests that the absence of mRNA decay results in defective transcriptional reprogramming that is reflected in the translating pool.

GO analyses of all DEGs in the mutant compared with WT after shift to host temperature further confirmed enrichment of genes involved in translation and protein folding in the mutant (Fig. 4d). Functionally related classes of genes underrepresented in the mutant-included cell division, DNA repair, and DNA replication. A shockingly large amount of genes involved in transcription were also downregulated compared with wild-type, further suggesting that mRNA decay is required for transcriptional regulation in response to stress.

**Translatome reprogramming precedes transcriptome remodeling**. Given that there appears to be no translational repression of newly synthesized mRNAs in the ccr4Δ mutant, yet a major misregulation of the transcriptome is seen, we looked closer at the drivers of transcriptome reprogramming, transcription factors. The stress-activated translation of transcription factors may be required for transcriptome reprogramming during temperature stress, and a defect in transcription factor translation and activation could contribute to the defect in temperature adaptation in the ccr4Δ mutant. We cross-referenced our RNA-seq dataset with the C. neoformans transcription factor collection characterized previously[24], and identified 66 transcription factors that were downregulated in the transcriptome of the ccr4Δ mutant at host temperature (Supplementary Dataset 2). Of these, 58 were also found downregulated in the polysome-associated fractions. The downregulation of three of these transcription factors were verified by qRT-PCR (Supplementary Table 1). Downregulated translation of these factors may explain the misregulation of the transcriptome in the ccr4Δ mutant.

**mRNA decay is required for cell wall remodeling at 37 °C**. Although they did not appear in our GO analysis, a manual curation of genes significantly repressed in the mutant revealed several genes involved in cell wall remodeling (Supplementary Table 2). We utilized qRT-PCR to validate differential expression of four of these transcripts (Supplementary Table 1). It was previously determined that the ccr4Δ mutant has irregular cell wall chitin deposition and displays sensitivity to cell wall-perturbing agents[17].

C. neoformans is unique among the fungi in that most cell wall glucans are masked, allowing C. neoformans to evade detection by glucan-recognizing pattern recognition receptors. Because many of the cell wall genes in our data set were glucan-modifying enzymes, we assessed glucan exposure in the ccr4Δ mutant. We utilized antibodies specific for α-(1,3)-glucan or β-(1,3)-glucan to stain cells of the wild-type or ccr4Δ mutant grown at 30 °C or 37 °C for exposed glucans by immunofluorescence microscopy and flow cytometry. While fluorescence was undetectable above background levels in the wild-type at either temperature, we observed large areas of exposed β-(1,3)-glucan on a sub-population of ccr4Δ cells only when grown at 37 °C (Fig. 5). Notably, we did detect α-(1,3)-glucan on ccr4Δ cells grown at 30 °C and 37 °C. The large crescent-shaped staining pattern was reminiscent of our previous studies that demonstrated β-(1,6)-glucan unmasking in the ccr4Δ mutant at 37 °C[22]. Because this staining pattern occurs reproducibly on a sub-population of cells, we asked whether the exposure was occurring on cells that were original to the shifted population, or on daughter cells that emerged during exposure to stress. We stained the original population of cells covalently with Alexafluor-488 and incubated the cells at 37 °C followed by β-(1,3)-glucan staining. Areas of glucan exposure occurred in daughter cells (cells lacking the alexafluor stain) of irregular morphology providing an explanation for the presence of exposed glucans in only a sub-population (Supplementary Fig. 5). Together, our results suggest that Ccr4-mediated decay contributes to cell wall glucan masking during host-temperature stress.

**Cell wall structure is disordered at 37 °C in C. amylolentus**. When we examined the wild-type C. neoformans strain for detection of α-(1,3)- and β-(1,3)-glucans, we did not observe either at 30 °C or at 37 °C, suggesting that cell wall remodeling masks these glucans during host-temperature stress. Given that RP transcripts are not downregulated upon host-temperature stress in C. amylolentus, we were curious to determine if downstream consequences of defective mRNA decay that we observed in the C. neoformans ccr4Δ mutant may also occur in the non-pathogen. Notably, C. amylolentus is an understudied organism and information regarding the cell wall structure and composition is lacking. While α-(1,3)-glucan was not significantly detected in C. amylolentus (Fig. 5a), β-(1,3)-glucan was exposed only at 37 °C in C. amylolentus, but was masked at 30 °C like the C. neoformans ccr4Δ mutant (Fig. 5b).

**Ccr4-mediated glucan masking aids in immune evasion**. The C-type lectin-like receptor Dectin-1 recognizes β-(1,3)-glucan and

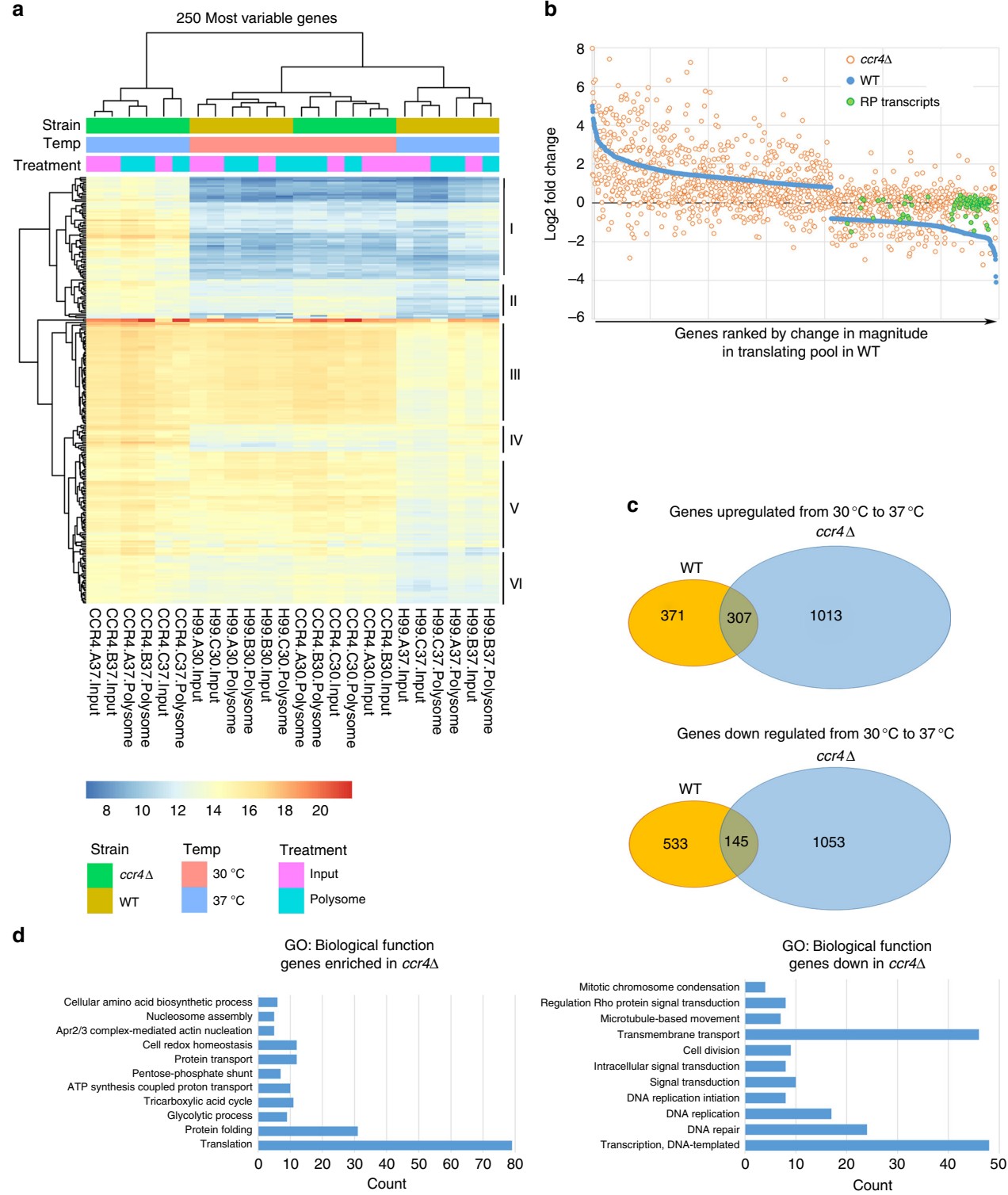

**Fig. 4** mRNA decay is required for reprogramming the translatome and transcriptome. **a** Heatmap of expression levels for the top 250 most variable genes in our total data set. The color scale defines the measured expression of a given gene. **b** Genes with significant change in the polysomal fractions following a shift from 30 °C to 37 °C in the wild-type identified by RNA-Seq were plotted in order of log2 magnitude change (blue circles). The magnitude of change for each corresponding gene in the *ccr4Δ* mutant was overlaid (orange circles). RP genes are highlighted in green. **c** Venn diagrams demonstrate genes that are upregulated and downregulated in response to host temperature in WT and the *ccr4Δ* mutant following growth at 37 °C for 1 h. **d** GO analyses for genes differentially expressed in the *ccr4Δ* mutant compared with wild-type 1 h after a shift to host temperature

plays a key role in regulating infection by several fungi (reviewed in ref. [25]). Interestingly, a previous study demonstrated that Dectin-1 was not required for protective responses in mice following intratracheal infection with *C. neoformans*[26]. It is likely that the cryptococcal capsule combined with strategically regulated cell wall remodeling keeps cell wall glucans hidden from Dectin-1 recognition. It was previously reported that capsule size and porosity is not affected in the *ccr4Δ* mutant[17]. To determine

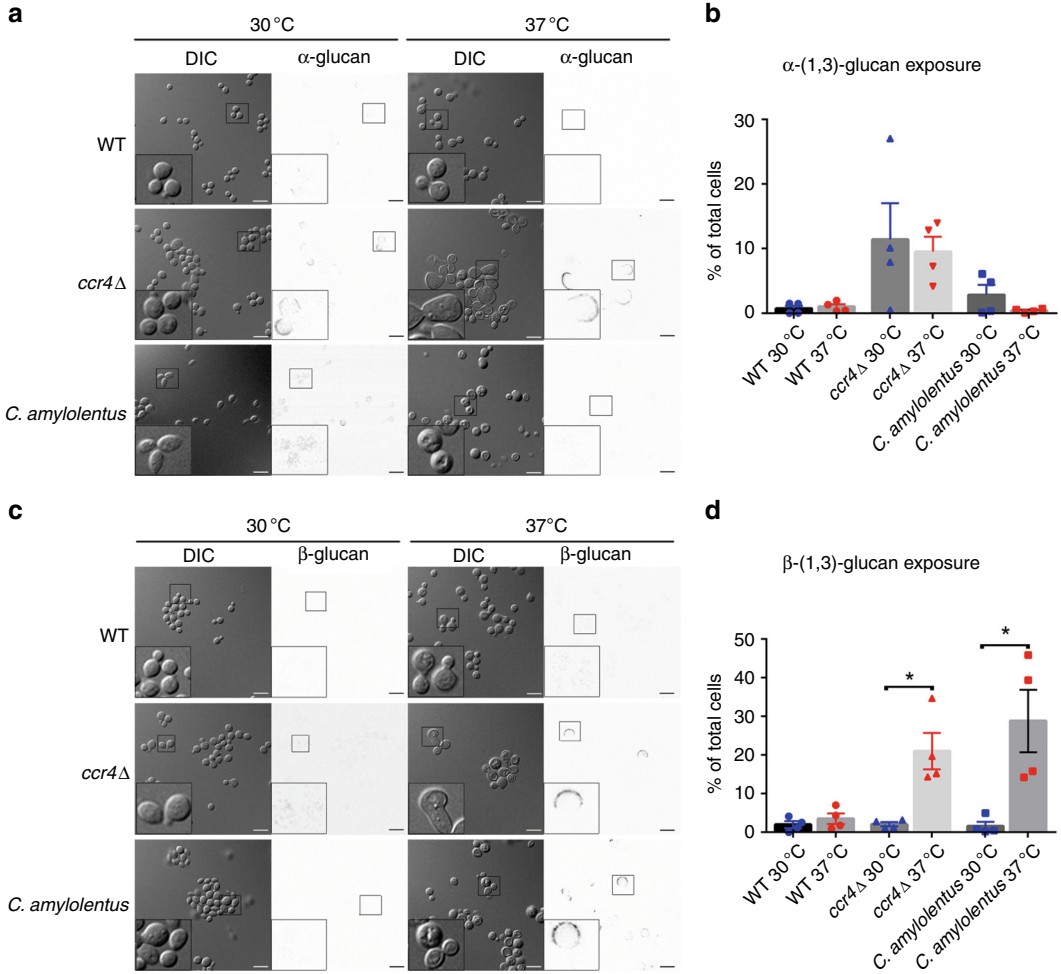

**Fig. 5** mRNA decay prevents unmasking of cell wall glucans at 37 °C. Wild-type *C. neoformans*, the *ccr4Δ* mutant, and wild-type *C. amylolentus* were grown at 30 °C or 37 °C and stained with α-(1,3)-glucan antibody MOPC104e (**a**, **b**) or anti-β-(1,3)-glucan antibody (**c**, **d**). Cells were visualized by fluorescence microscopy (**a**, **c**). Scale bar = 10 μm. Insets show magnified views of cells outlined in boxes. Glucan exposure was measured by flow cytometry (**b**, **d**). Graphs are representative of the mean percentage of cells ± s.e.m. (*n* = 4) exposing α-(1,3)-glucan (**b**) or β-(1,3)-glucan (**d**), and significance was determined by Kruskal–Wallis and Dunn's test post hoc. Source data are provided as a Source Data file

if Dectin-1 can recognize the exposed glucan on the *ccr4Δ* mutant, cells grown at 30 °C or 37 °C were incubated with a mouse Dectin-1:human-Fc conjugate, and binding was detected by immunofluorescence microscopy and flow cytometry. Dectin-1 binding was not detectable on wild-type cells at either temperature, but was evident by both methods for a population of *ccr4Δ* cells grown at 37 °C (Fig. 6a, b).

We also incubated the thermointolerant, non-pathogen, *C. amylolentus*, with the Dectin1-Fc conjugate followed by immunofluorescent labeling. Similar to the *C. neoformans ccr4Δ* mutant Dectin1-Fc-bound *C. amylolentus* only when cells were grown at 37 °C (Fig. 6a, b).

To determine if mRNA decay-regulated cell wall remodeling and glucan masking are important for evading the resident macrophages within a host pulmonary environment, we fluorescently labeled wild-type and *ccr4Δ* cells, infected mice intranasally, and assessed phagocytosis by macrophages by flow cytometry. Analyses of both the bronchoalveolar lavage (BAL) and digested lungs demonstrate significantly higher frequency of macrophages that engulfed *ccr4Δ* cells than wild-type cells (Fig. 6c). Our results suggest that exposure of glucans typically masked in the wild-type allows for recognition of the *ccr4Δ* mutant by host pattern recognition receptors.

**Hob1 regulates β-glucan masking during 37 °C stress.** Given the repression of transcription factors in the *ccr4Δ* mutant, we set out to determine if any of them are regulators of unmasking. We screened 24 transcription factors from the Bahn collection that were reported to exhibit either cell integrity stress sensitivity or thermosensitivity[24] for anti-β-(1,3)-glucan binding by flow cytometry. We identified a single transcription factor mutant, *hob1Δ*, that exhibited β-(1,3)-glucan exposure in our assay (Fig. 7). Based on our RNA-seq data set, *HOB1* is repressed 6.3-fold in the transcriptome, and 5.3-fold in the translatome in the *ccr4Δ* mutant at 37 °C (Supplementary Dataset 2). Validation of the seq data for *HOB1* demonstrated that is was indeed repressed in the *ccr4Δ* mutant (Supplementary Table 1).

These results suggest a model whereby *C. neoformans* adapts to host temperature and masks β-1,3-glucan by the rapid degradation of mRNAs encoding ribosomal proteins, allowing for the rapid, stress-responsive translation of transcription factors, promoting transcriptional reprogramming in response to temperature stress. Among these transcription factors, many are involved in thermotolerance, stress tolerance, and pathogenesis. Thus a major role of mRNA decay is to permit stress-responsive translation of transcription factors and their targets required for temperature adaptation and antigen masking.

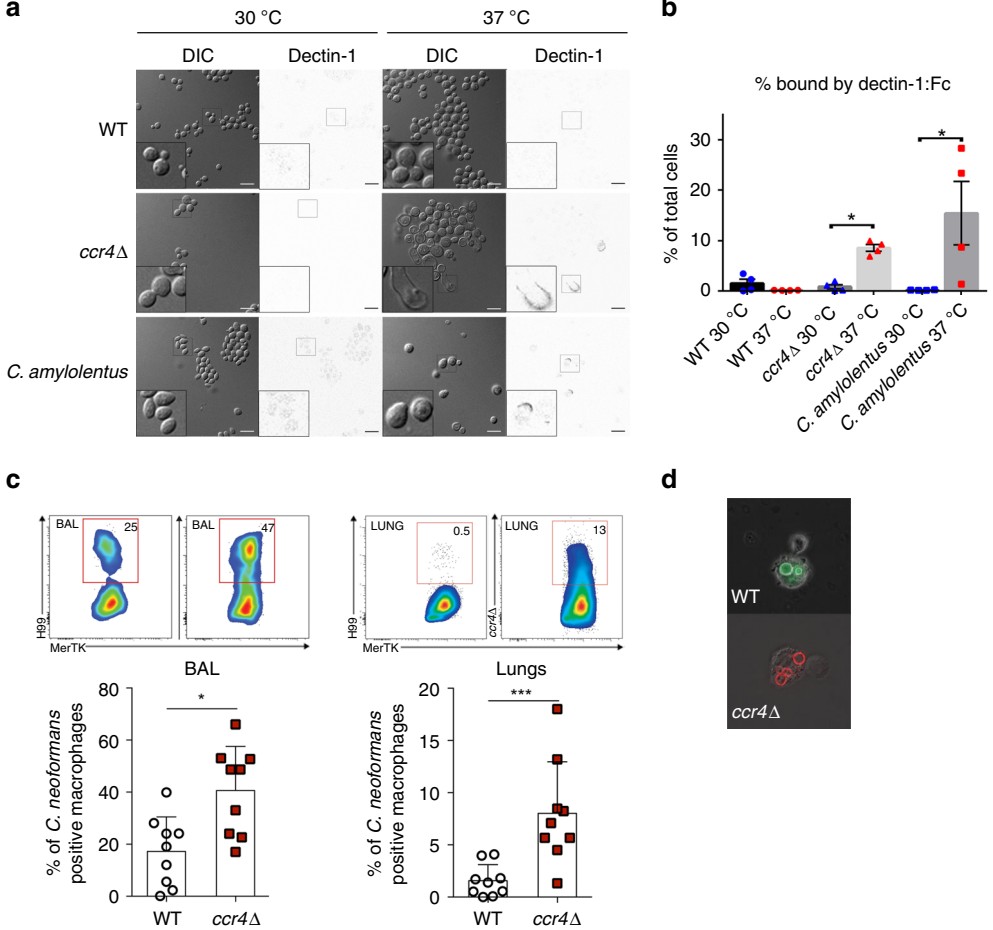

**Fig. 6** Glucan exposure results in Dectin-1 recognition and phagocytosis. **a** Wild-type *C. neoformans*, the *ccr4Δ* mutant, and wild-type *C. amylolentus* were grown at 30 °C or 37 °C and incubated with mouse Dectin-1-human-Fc protein conjugate followed by fluorescent labeling and microscopy. Scale bar = 10 μm. Insets show magnified views of cells outlined in boxes. **b** Recognition of cells by Dectin-1 was measured by flow cytometry. Graph depicts the mean percentage of cells ± s.e.m. (*n* = 4) bound by mouse Dectin-1:human-Fc conjugate, and significance was determined by Kruskal–Wallis and Dunn's test post hoc. **c** BalbC/J mice were infected intranasally with 2.5 × 10⁶ Alexafluor-488-labeled WT cells or Alexafluor-594 labeled *ccr4Δ* cells for 5 h. BAL and digested lung were analyzed by flow cytometry for macrophages associated with *C. neoformans*. Graphs show the mean percentage of macrophages containing cryptococci ± SD (*n* = 3), and significance was determined by Wilcoxen–Mann *U* test. Gating strategy is demonstrated in Supplementary Fig. 6. **d** Fluorescence microscopy images from BAL samples showing WT (green) or *ccr4Δ* (red) cells inside phagocytes. Source data are provided as a Source Data file

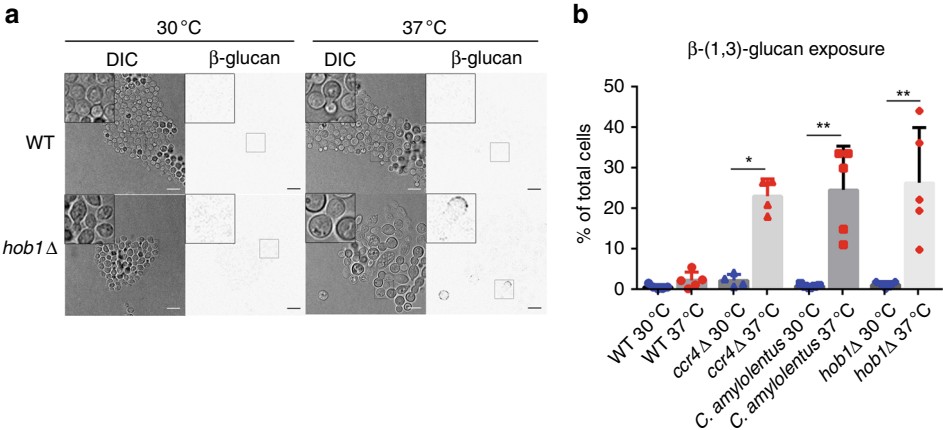

**Fig. 7** The transcription factor, Hob1, contributes to cell wall masking at 37 °C. **a** Wild-type *C. neoformans*, the *ccr4Δ* mutant, the wild-type *C. amylolentus*, and the *C. neoformans hob1Δ* mutant strains were grown at 30 °C or 37 °C and stained with β-(1,3)-glucan antibody for fluorescence microscopy. Scale bar = 10 μm. Insets show magnified views cells outlined in boxes. Glucan exposure was measured by flow cytometry (**b**). Graphs are representative of the mean percentage of cells ± s.e.m. (*n* = 5) exposing β-(1,3)-glucan, and significance was determined by Kruskal–Wallis and Dunn's test post hoc. Source data are provided as a Source Data file

## Discussion

Environmental pathogens must be equipped to adapt to the host environment, and the host must be equipped to recognize and neutralize the pathogen. In this study, we demonstrate that adaptation to host temperature and immune evasion are linked, as both require transcriptome and translatome reprogramming via deadenylation-dependent mRNA decay. Using the non-pathogen *C. amylolentus* as a comparator, we further demonstrate the link between mRNA decay, temperature adaptation, and antigen masking, as *C. amylolentus* behaves similarly to the decay-defective *C. neoformans ccr4Δ* mutant.

Our work with *C. amylolentus* suggests that the immediate and specific destabilization of RP transcripts is critical for adaptation to host-temperature stress and is a prerequisite for maintained growth in this condition, and thus required for persistence in the human host. Further work is needed to define the molecular mechanisms that specify decay of RP transcripts, and the upstream signaling pathways that communicate temperature stress to the decay machinery. With the recent sequencing of the *C. amylolentus* genome, our future endeavors include assessing differences compared with *C. neoformans* that may shed light on evolutionary changes that have conferred thermotolerance.

To better understand the importance and purpose of RP repression, we utilized RNA-seq to determine the consequences of stabilized RP transcripts. In *S. cerevisiae*, mRNA degradation promotes removal of target transcripts from the translational machinery[23,27] and allows for faster repression than does repressed transcription, and in the event of sudden stress, a fast response is needed[28]. Consistent with these reports, our data demonstrate that Ccr4-mediated decay removes RP transcripts from the translating pool, maintains a cellular balance of mRNA levels and contributes to overall gene expression changes that promote host-temperature adaptation.

The transcriptome remodeling defect in our mutant suggests that reprogramming of the translatome may be required for appropriate transcriptional control in response to host temperature. We identified several genes that encode transcription factors that were significantly repressed in the translatome and transcriptome of the decay mutant at 37 °C. Transcription factors are often self-regulating. Thus, failure of these genes to be adequately translated due to competition with stabilized RP mRNAs could result in their repressed transcription. While our data demonstrated that newly synthesized transcripts enter the translating pool in the mutant, ribosome occupancy could not be determined by our experimental design thus, it is possible that rates of translation of these factors may be decreased.

Changes in cell wall composition have previously been reported in *C. neoformans* in response to stresses encountered in the host environment, and mutations in genes that effect synthesis and/or localization of cell wall components often result in host-like stress sensitivity and attenuated virulence[29–31]. Cell wall integrity (CWI) is specifically important for growth at elevated temperature as mutants with cell wall defects often have slowed growth or no growth at 37 °C unless grown in the presence of an osmostabilizing agent. We revealed that cell wall remodeling keeps glucans masked following host-temperature stress. Significant repression of genes involved in cell wall remodeling in the mutant suggests that a transcriptional regulator(s) of these genes is not activated or translated, or is the result of transcriptional repression. We identified 25 transcription factors that are repressed in the *ccr4Δ* mutant that could potentially regulate cell wall remodeling. Furthermore, we demonstrated that Hob1 may specifically be a key regulator in host-temperature-induced masking of β-(1,3)-glucan. Future assessment of promoters bound by Hob1 and gene expression effected by the absence of Hob1 will elucidate its role in cell wall remodeling. That

*C. amylolentus* lacks enhanced RP transcript decay and exposes β-glucan in response to host temperature, like the *C. neoformans ccr4Δ* mutant, further suggests that regulated mRNA decay in response to host temperature is a key attribute for adaptation. Although BLAST analysis identifies a putative homolog of Hob1 in *C. amylolentus*, it is currently unknown whether the downstream signaling of mRNA decay is conserved in the non-pathogen or if the Hob1 homolog functions in a similar manner. Further investigation is needed to identify both upstream and downstream signaling components to understand Ccr4-related signaling cascades and determine if these components are present and conserved in *C.amylolentus*.

Dectin-1 is the pattern-recognition receptor on host macrophages associated with recognition of fungal cell wall β-(1,3)-glucan and contributes to recognition and clearance of several pathogenic fungi, including *Candida albicans*, *Pneumocystis carinii*, and *Aspergillus fumigatus*[32–34], but is not required for a protective response to *C. neoformans* infection[26]. In addition, wild-type *C. neoformans* is not bound by serum mannan-binding lectin (MBL)[17], and it was reported that human MBL deficiency is not correlated with susceptibility to cryptococcosis[35]. That wild-type cells are able to remodel and mask their cell wall in response to host-induced stress likely explains why such pattern recognition receptors are dispensable for protection.

A recent study demonstrated that a strain lacking the novel cell wall regulatory protein "Macrophage-activating cell wall regulator 1", mislocalizes the major beta-glucan synthase and exposes cell wall chitooligomers and glucans in response to host-like pH and glucose deprivation, resulting in more efficient phagocytosis and amplified production of tumor necrosis factor-α (TNFα) partially dependent on Dectin-1[30]. In addition, an acapsular strain of *C. neoformans* demonstrated beta-glucan receptor-mediated phagocytosis resulting in production of tumor necrosis factor alpha (TNF-α) and granulocyte–macrophage-colony-stimulating factor (GM-CSF), which enhanced phagocytosis of encapsulated *C. neoformans*[36]. We showed here that a population of *ccr4Δ* mutant cells expose glucans and are recognized by Dectin-1 when grown at host temperature, likely explaining increased phagocytosis of mutant cells by lung macrophages.

*C. neoformans* is both an environmental fungus and a successful pathogen of humans. Its success can be attributed to both adaptability and immune evasion. In this study, we demonstrate that these two pathogenic attributes are linked to the most central process in living cells, translation. More specifically, loss of mRNA decay not only locks degradation-destined mRNAs into the translating pool but also blunts the ability of the pathogen to reprogram aspects of its transcriptome, further impairing its stress responsiveness. Our analysis revealed that cell wall remodeling requires decay-dependent reprogramming of the translatome, and subsequently, the transcriptome. The abrogation of this reprogramming reveals a link between adaptation and immune evasion, and identifies a central process, mRNA decay, that governs both important pathogenic attributes of this environmental human pathogen.

## Methods

**Yeast cells**. C . neoformans strains H99 (wild-type), *ccr4Δ*, *C. amylolentusCCR4::ccr4Δ*, *hob1Δ* and *C. amylolentus* strain CBS 6039 were long-term stored at −80 °C in glycerol stocks. Strains were streaked out and grown on Yeast Peptone Dextrose (YPD) agar plates, stored at 4 °C, and used to prepare overnight starter cultures. Overnight cultures were used to seed fresh cultures, and cells were typically grown for ~5 h in baffled flasks to mid-log phase in YPD at 30 °C, 250 rpm, for use in experiments.

The *ccr4Δ* mutant and *C. neoformans CCR4::ccr4Δ* were constructed as described previously[17]. The *hob1Δ* mutant was obtained from the Bahn transcription factor knockout collection purchased from the Fungal Genetics Stock Center, and *C. amylolentus* was obtained from ATCC.

To construct the *C. amylolentus*CCR4::*ccr4Δ* strain, the *CCR4* coding sequence was PCR amplified from *C. amylolentus* genomic DNA template with MunI and XbaI ends using primers F-CamyloCCR4 and R-CamyloCCR4 (Supplementary Table 3). The 1-kb upstream region of CCR4 was amplified from *C. neoformans* genomic DNA template with KpnI and MunI ends using primer F-CneoCCR4up and R-CneoCCR4up (Supplementary Table 3). The 1-kb downstream region of CCR4 was amplified from *C. neoformans* genomic DNA template with XbaI and KpnI ends using primers F-CneoCCR4down and R-CneoCCR4down. All PCR amplicons were digested with respective restriction digest enzymes, purified, and ligated into a pBluescript plasmid that contains the G418 resistance cassette inserted at the XhoI site. The ligated plasmid was transformed into electrocompetent DH10 cells. Purified plasmid was precipitated onto gold microbeads and randomly integrated into the *C. neoformans ccr4Δ* mutant strain by biolistic transformation[37]. Genomic integration was verified by southern blot, and expression was confirmed by northern blot.

**Mice.** Balb/cJ mice were purchased from Jackson Laboratories. All mice were female, aged 5–6 weeks. Mice were randomly housed 2–3 per cage in the BSL2 suite in the Laboratory Animal Facility at the University at Buffalo. All experiments involving mice were performed in compliance with all ethical regulations for animal testing and research, and were approved by IACUC at SUNY University at Buffalo.

***RPL2* steady-state and stability assays.** Cells were grown at 30 °C in YPD. Once mid-log phase was reached, cells were pelleted by centrifugation at 4000 rpm for 5 min. For steady-state analyses, the pellet was resuspended in pre-warmed 37 °C YPD, and 5-ml aliquots were pelleted every 30 min for 3 h. For stability analyses, cell pellets were resuspended in pre-warmed YPD supplemented with 250 μg mL$^{-1}$ of the translational inhibitor 1,10-phenanthroline (Sigma), and 5-mL aliquots of each culture were pelleted every 15 min for 1 h. RNA was extracted from each sample using the RNeasy Mini Kit (Qiagen), and 3 μg of RNA for each sample was electrophoretically separated through an agarose-formaldehyde gel and then transferred to a nylon membrane. *RPL2* transcript was detected by hybridization with a P32-labeled DNA probe, followed by phosphor imaging. Primers used to amplify *RPL2* can be found in Supplementary Table 3. Hybridization signal was quantitated using Image Lab software and normalized to the fluorescence signal of the ribosomal bands. All uncropped images of gels and scanned blots can be found in the source data.

**Spot-dilution assay.** Cells were grown in YPD at 30 °C overnight, pelleted by centrifugation at 4000 rpm for 5 min, washed twice with sterile deionized water (SDW), and resuspended in SDW to an $OD_{600} = 1.0$. Five tenfold serial dilutions were prepared in SDW, and 5 μl of each dilution was spotted onto a YPD agar plate. Plates were incubated for 3 days at indicated temperatures.

**Growth curve analysis.** Cells were grown to mid-log phase in YPD at 30 °C. Cultures were adjusted to an $OD_{600} = 0.1$, and 100 μl were deposited into a 96-well plate in duplicate. Plates were incubated at 37 °C with orbital shaking in a BioTek plate reader, and $OD_{600}$ was recorded every 15 min for 15 h using Gen5 2.05 software.

**mRNA abundance assay.** Wild-type and *ccr4Δ* cells were grown to mid-log phase in YPD at 30 °C, 250 rpm. Half of the culture was resuspended in pre-warmed 37 °C YPD, and incubated at 37 °C for 1 h. Cells were pelleted, and RNA was extracted using RNeasy mini column kit (Qiagen). The total RNA was quantified, and equivalent amounts of RNA suspended in equivalent volume of RNase-free water were vacuum filtered through a slot blot manifold onto nylon membrane. A P32 end-labeled oligo-dT probe was used to assess mRNA abundance by northern blot.

**Polysome profiling.** For polysome analyses, cells were grown to mid-log phase in YPD. Half of the culture was pelleted, resuspended in pre-warmed 37 °C YPD, and incubated at 37 °C for 1 h. Cycloheximide was added to cultures (0.1 mg ml$^{-1}$), and cells were pelleted by centrifugation at 4000 rpm, 4 °C, for 5 min. Pellets were washed in polysome lysis buffer (20 mM Tris HCl, pH 8.0, 140 mM KCl, 5 mM MgCl$_2$, 1% Triton X-100, 25 mg ml$^{-1}$ heparin sodium sulfate, 0.1 mg ml$^{-1}$ cyclo-heximide) and pelleted as before. Supernatant was removed, and pellets were frozen in liquid nitrogen and stored at −80 °C until lysis. Pellets were thawed on ice, resuspended in 200 μl of lysis buffer, and transferred to 15 ml round-bottom tubes. An equal volume of glass beads was added to each tube, and cells were lysed mechanically by vortexing for 20 s, followed by incubation on ice for 40 s five times. Supernatant was transferred to a cold microfuge tube and centrifuged for 5 min, at 14,000 rpm, 4 °C. Cleared lysates were quantitated for RNA, and 250 μg was loaded on top of 10–50% sucrose gradients. Gradients were subjected to ultracentrifugation for 2 h, 39,000 rpm, 4 °C. Sucrose gradients were then pushed through a flow cell, and RNA was detected by UV–vis A254 absorbance. 16-drop fractions were simultaneously collected while the profile was recorded. Three fraction volumes of 100% ethanol were added to fractions, and samples were stored at −80 °C. Samples were centrifuged at 14,000 rpm for 5 min at 4 °C. Ethanol was removed, and pellets were resuspended in 250 μL of RNase-free water followed by the immediate addition of 750 μL of TRIzol LS (Invitrogen, ThermoFisher Scientific). RNA

extraction was performed according to the manufacturer's protocol, and RNA pellets were resuspended in 50 μL of RNase-free water. In instances when RNA distribution throughout the polysome profile was assessed, 10 μL for each fraction was run on a 1% formaldehyde-agarose gel, transferred to a nylon membrane, probed using a P32-labeled *RPL2* DNA probe, and detected by phosphor imaging.

**RNA sequencing.** TRIzol extracted RNA isolated from polysome fractions was pooled. Pooled polysomal RNA and RNA isolated from the inputs were separately purified by passage through RNeasy Mini columns. To remove rRNA, poly-adenylated mRNA was isolated from 4 μg of the total RNA using NEBNext poly(A) mRNA Magnetic Isolation Module (New England Biolabs) as per the manu-facturer's protocol, and RNA-seq libraries were constructed using NEBNext mRNA Library Prep Master Mix Set for Illumina (New England Biolabs) according to the manufacturer's protocol. RNA sequencing was performed on an Illumina HiSeq 2500 platform.

Basecalling files were converted to FASTQ format using illumina's bcl2fastq version 2.17.1.14 using default parameters. The quality of the sequencing was reviewed using FastQC version 0.11.5 and Fastq_Screen version 0.11.1. The resulting reads were trimmed for adapter sequence using cutadapt version 1.16. Alignments were performed using hisat2 version 2.1.0 against the Cryptococcus_neoformans_var._grubii_H99 reference genome and gene annotation set retrieved from FungiDB. Sequence alignments were compressed and sorted into binary alignment map (BAM) files using samtools version 1.7. Quantification of mapped reads was performed using Subread featureCounts version 1.6.0. Differentially expressed genes were detected using the Bioconductor package DESeq2 version 1.20.0.

For comparisons between strains or conditions, differentially expressed genes were defined as those with padj values < 0.05 and a fold change >± 1.75.

For GO analyses, DEGs were uploaded to and analyzed by DAVID 6.8 software using the BP_Direct filter. Only terms with p-values < 0.05 were included.

**Cell wall glucan detection and Dectin-1 recognition.** To visualize alpha-(1, 3)-glucan, beta-(1,3)-glucan, and binding by Dectin-1, ~2.5 × 10$^6$ cells were incubated with either 15 μg ml$^{-1}$ alpha-glucan-specific antibody MOPC-104E (IgM; Sigma), 15 μg ml$^{-1}$ anti-beta-(1,3)-glucan antibody (IgG; Australia Biosupplies), or 5 μg ml$^{-1}$ of mouse Dectin1-human-Fc conjugate (IgG, Adipogen) in 250 μL volume for 1 h at 37 °C with rotation. Cells were washed three times with PBS, and then incubated with 10 μg ml$^{-1}$ Alexafluor-488-conjugated anti-mouse IgM (Invitrogen) for alpha-glucan, Alexafluor-488-conjugated anti-mouse IgG (Invitrogen) for beta-glucan, or Alexafluor-488-conjugated anti-human IgG (Invitrogen) for Dectin1 recognition for 30 min at 37 °C with rotation. Cells were washed three times in PBS, fixed in 3.7% formaldehyde for 20 min, and washed and resuspended in PBS. Cells were viewed using a Leica TCS SP8 Confocal Microscope. Flow cytometry data were acquired using a BD LSRFortessa Cell Analyzer, and analyzed using FlowJo version 10.0.8 (Tree Star). Experiments were performed four times.

For detection of beta-(1,3)-glucan on newly emerging daughter cells during stress, *ccr4Δ* cells grown in YPD at 30 °C were washed and resuspended in 1 mL PBS, pH 8.0. Cells were stained with Alexafluor-488 by adding them to one vial of dye provided in the Alexafluor-594 protein labeling kit (Invitrogen). After 30 min of incubation and stirring, cells were washed three times with PBS to remove residual dye, resuspended in YPD and incubated at 37 °C overnight. Cells were then stained for beta-(1,3)-glucan as above, except Alexfluor-594-conjugated anti-mouse IgG secondary antibody was used.

**Mice infection and lung macrophage phagocytosis assay.** Alexaflour labeling of cryptococcal cells was performed as described previously[38]. To fluorescently label cells, 1 mL of overnight culture was washed with 5 mL of PBS, pelleted and resuspended in 1 mL of PBS, and pH was adjusted to 8.0 by the addition of 50 μL of sodium bicarbonate. The cell suspension was added to one vial provided in the Alexafluor-488 (WT) or Alexaflour-594 (*ccr4Δ*) protein labeling kit (Invitrogen), and cells were stained for 30 min, stirring on a magnetic stir plate. Cells were washed three times, and resuspended in PBS. Following the labeling procedure, cells were counted by the hemacytometer and adjusted to 1 × 10$^8$ cells mL$^{-1}$ in PBS. Each inoculum was serially diluted and plated to confirm that mice were infected with equivalent cellular concentrations. For each strain tested, three 5–6-week-old female BalbC/J mice (Jackson Laboratories) were anesthetized with isofluorane and intranasally infected with 25 μL (2.5 × 10$^6$ cells) of inoculum. Mice were killed via CO$_2$ 5 h post infection. To obtain BAL, lungs were lavaged with 1.5 mL sterile PBS three times using a 22-gauge blunt-end needle placed in the trachea. Red blood cells were lysed in ACK lysing buffer (Lonza) twice, and BAL was resuspended in 10% media (RPMI with 10% FBS, 1% penicillin–streptomycin, 1 mM sodium pyruvate, 0.1% β-mercaptoethanol, 25 mM HEPES) for a single-cell suspension.

For isolation of lung tissue leukocytes, mice were subsequently transcardially perfused with PBS. Lungs were harvested and minced in digestion media (RPMI, 1% penicillin–streptomycin, 1 mM sodium pyruvate, 0.1% β-mercaptoethanol, 25 mM HEPES, and 150 μg ml$^{-1}$ Dnase I [Sigma-Aldrich], 50 μg ml$^{-1}$ liberase TL [Roche]). Tissues were digested at 37 °C for 45 min and sequentially passed through a 70-μm and 40-μm filter. Red blood cells were lysed in ACK lysing buffer, and mononuclear cells were resuspended in 10% media for a single-cell suspension.

For extracellular and intracellular flow-cytometric analysis of tissue lymphocytes, single-cell suspensions were stained with HBSS containing extracellular antibody stains and Live/Dead Fixable Aqua dead cell stain (Life Technologies). For identification of macrophages, samples were stained with anti-CD11b-BV605 (BD Horizon, clone M1/70), anti-CD45-APC-eFluor 780 (ebioscience, clone HI30), anti-MerTK (Biolegend, clone 2B10C45), anti-CD11c (eBioscience, clone N418). Samples were washed and resuspended in flow cytometry buffer/FACS Buffer (PBS, 1% bovine serum albumin (Sigma-Aldrich), 2 mM EDTA (Life Technologies)) for acquisition. Flow-cytometry data were acquired using a BD LSRFortessa Cell Analyzer and analyzed using FlowJo version 10.0.8 (Tree Star).

**qRT-PCR.** For detection of cell wall-remodeling genes, polyadenylated mRNA was isolated from 5 μg of total RNA using NEBNext poly(A) mRNA isolation module (New England BioLabs) and cDNA was produced using iScript cDNA synthesis kit (Bio-Rad). For RT-PCR, cDNA was diluted 1:10 and added to IQ SYBR Green Supermix (Bio-Rad) per protocol instructions. Primers for cell wall-remodeling genes can be found in Supplementary Table 3. The data were acquired on a CFX Connect Real-Time PCR Detection System (Bio-Rad). A histone acetyl transferase (CNAG_03297) was used as an internal control, and negative control samples without reverse transcriptase were included.

For detection of transcription factor genes, RNA was isolated using an RNeasy mini kit (Qiagen) and DNase treated using Turbo DNase-free kit (ThermoFisher Scientific). For RT-PCR, cDNA was synthesized using the High Capacity cDNA reverse transcription kit with RNase Inhibitor (ThermoFisher Scientific), diluted 1:5 and added to qPCR SyGreen Blue Mix LoROX (PCR Biosystems) per protocol instructions. Primers for amplifying transcription factor genes can be found in Supplementary Table 3. The data were acquired on a CFX Connect Real-Time PCR Detection System (Bio-Rad). Mitofusin (CNAG_06688) was used as an internal control, and negative control samples without reverse transcriptase were included.

Two technical replicates and three biological replicates were performed for all reactions. The ΔΔCt method was used to calculate differences in expression.

**Quantification and statistical analysis.** Statistical analyses were performed using GraphPad Prism (version 6.05) software. For all analyses, significance was defined as follows: $*p < 0.05$; $**p < 0.01$; $***p < 0.001$ unless otherwise noted. Additional statistical information can be found in figure legends.

Statistical analyses for stability data were obtained by determining the least squares fit of one-phase exponential decay nonlinear regression. Significance between curves was detected by sum-of-squares F test, with $p < 0.05$ determining that the data fall on separate regression lines, and therefore exhibit different rates of decay.

Statistical analyses for expression of *RPL2* following shift to 37 °C were obtained by performing ANOVA with Dunnett's test post hoc comparing the mean of each time point to $t = 0$ for that strain.

For growth curve analyses, the doubling time for each strain was determined from a period of exponential growth. The doubling times of each strain from three biological replicates were compared by ANOVA and Tukey's test post hoc.

Statistical analysis to compare polyA mRNA abundance between wild-type and the *ccr4Δ* mutant was performed using ANOVA and Tukey's test post hoc.

For RNA sequencing, DESeq2 was used to determine differential expression using a negative binomial generalized linear models, dispersion estimates, and logarithmic fold changes. DESeq2 calculates log2 fold changes and Wald test *p*-values as well as performing independent filtering and adjusts for multiple testing using the Benjamini–Hochberg procedure to control the false discovery rate (FDR).

Statistical analyses comparing macrophage association of wild-type and the *ccr4Δ* mutant were performed using Wilcoxen–Mann *U* test.

For glucan exposure and Fc-Dectin-1 binding significant differences were determined using Kruskal–Wallis, and Dunn's test post hoc.

**Reporting summary.** Further information on research design is available in the Nature Research Reporting Summary linked to this article.

## Data availability

RNA sequencing data for *C. neoformans* WT and *ccr4Δ* have been deposited in the GEO under code GSE121183. All other relevant data are available from the corresponding author on request. Source data for Figs. 1–3, 5–7, Supplementary Figure 1, 2, and Supplementary Table 2 are provided with the paper.

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

## Acknowledgements

This work was supported by NIH R01AI131977 to J.C.P. The Bahn Transcription Factor Deletion Collection was obtained from the Fungal Genetics Stock Center (Manhattan, Kansas, USA).

## Author contributions

Conceptualization: A.L.B. and J.C.P.; methodology: A.L.B., R.M.J., and J.L.; formal analysis: A.L.B., J.B., D.Y. and R.M.J.; investigation: A.L.B. and R.J.M.; writing—original draft: A.L.B.; writing—review & editing: A.L.B., J.C.P., J.L., and E.A.W.; visualization: A.L.B.; supervision: J.C.P. and E.A.W.; resources: E.A.W. and J.C.P.

## Competing interests

The authors declare no competing interests.
