## [Peer Review File · Nature Communications]

Reviewers' comments:

Reviewer #1 (Remarks to the Author):

Previously the Panepinto group has reported that Ccr4-mediated mRNA decay of ribosomal protein (RP) transcripts is critical for the thermotolerance of *Cryptococcus neoformans*. In this manuscript, Bloom et al. further investigated its role by comparing thermo-intolerant *Cryptococcus amyloletus* and thermoresistant *C. neoformans*. They found that the high temperature-induced Ccr4-mediated RP transcript decay did not properly occur in *C. amyloletus*, suggesting that this phenomenon could be linked to the fungal pathogenicity. Subsequently, they performed RNAseq and polysome profiling assays for the wild type and *ccr4* mutants in *C. neoformans* and found that global transcriptome and translome profiles are significantly affected by *ccr4* mutation. Particularly, they demonstrated that Ccr4 regulates exposure of cell wall glucan and recognition by host Dectin-1, which could be associated with phagocytosis by macrophage, and cell cycle controls during host-temperature adaptation.

As a successive story of *C. neoformans* Ccr4 by this group, this manuscript provides an interesting piece of data for the role of Ccr4 in host temperature adaptation and the fungal pathogenicity by analyzing thermo-tolerant and thermos-intolerant *Cryptococcus* species. Nevertheless, they still failed to provide any mechanistic insight into how Ccr4 mediates host-temperature adaptation, cell wall remodeling, and cell cycle regulation occur. The multiple roles of the Ccr4-Not transcriptional complex in *S. cerevisiae* have been well reported, including deadenylation-dependent mRNA decay, nuclear-transcribed mRNA poly(A) tail shortening, replication fork protection, DNA replication, and regulation of transcription and transcription elongation. Therefore, at this stage, one would wonder what are the common and divergent roles of Ccr4 in *C. neoformans*, compared with non-pathogenic yeasts, and its regulatory mechanisms in detail. But this manuscript still seems rather descriptive for the role of Ccr4 in this sense.

In this study, the authors demonstrated that the high temperature-induced Ccr4-mediated RP transcript decay did not properly occur in *C. amyloletus* and proposed that this phenomenon could be linked to the fungal pathogenicity. Nevertheless, they did not experimentally prove that these two phenotypic traits are actually linked. If the RP transcript decay properly occurs in *C. amyloletus*, will this non-pathogenic fungus become as tolerant to high temperature and pathogenic as *C. neoformans*? Furthermore, it is not clear whether the RP transcript decay difference between *C. neoformans* and *C. amyloletus* results from different functions of Ccr4 orthologs per se or other regulators upstream of Ccr4. The former hypothesis could be proven by allele swapping experiment. The latter hypothesis could be proven only after the detailed mechanistic details of the Ccr4-mediated signaling pathway is elucidated.

I have the following other comments.

In Figure 1 and other figures containing northern blot data, the authors need to quantitatively measure gene expression levels by using different loading controls. rRNAs are not appropriate for quantitative measure as they are too abundant (about 80% of total RNA) and their signals are too strong to be used as quantitative loading controls. Instead actin, tubulin or other housekeeping genes could be used. Otherwise, they should provide quantitative RT-PCR data with three biological independent replicates and perform statistical analysis to determine whether any difference is meaningful or not. Currently, the authors did not perform any statistical analysis. Based on the Figure 1a northern blot, there seems to be a difference in transcript levels between zero time and 3 hr in *C. amyloletus*. In addition, the label "WT" should be written as "*C. neoformans*", because *C. amyloletus* is also WT in the given species.

Figure 2a is hard to understand. What do these graphs mean? There is no definition for Y- and X-axes. There is no labeling on each lane of northern blot data. Based on the rRNA loading controls, they are not equally loaded. In fact, Figure 2a was described later than Figure 2b in the main text.

Also Figure 2C is described later than Figure 3a. So in general the order of figures should be rearranged according to the description order in the main text.

The quality of Figure 4a, 4c and 5a is too low. The colored fluorescence images could be replaced with better ones. For example, the result of Figure 5a does not correspond to the Figure 5b. In the Figure 5a, the fluorescence signals bound by Dectin-1 in *C. amyloletus* at 30C appear to be identical to those at 37C, which does not correspond to the result of Figure 5b.

The graph of Figure 6c requires statistical analysis data to indicate which samples are significantly different.

Page 12, line 1: Figure 5b should be Figure 5c.

On page 12, the authors mentioned that the *ccr4* mutant shows cell cycle defects following host temperature stress. Based on the figure 6a, the *ccr4* mutant is already defective in cell cycle regulation even under unstressed conditions (30C). So the role of Ccr4 in cell cycle regulation is not a high-temperature specific phenotypic trait.

Reviewer #2 (Remarks to the Author):

The authors build upon their prior work in post-transcriptional control of mRNA abundance by pursuing studies in the role of the Ccr4 protein in microbial pathogenesis using the fungal pathogen *Cryptococcus neoformans* as a model system. They document mRNA decay of the ribosomal protein (RP) gene RPL2 in response to elevated temperature. Stress-induced inhibition of RP protein transcripts is well documented in other species, likely redirecting the translational machinery away from translating these highly abundant proteins and toward new stress-response proteins. Interestingly, they demonstrate that this mRNA decay response is not observed in a related *Cryptococcus* species that is unable to grow at elevated temperature, suggesting an association between temperature-regulated RP mRNA stability and thermal intolerance. They offer a nice control that RP transcript decay was observed in both *Cryptococcus* species upon carbon starvation.

Much of the paper is a further analysis of the role of Ccr4 in mRNA degradation. Total mRNA levels are stable in WT cells in response to a temperature shift, but these levels are elevated in the *ccr4* mutant strain at 37C, suggesting a defect in mRNA decay, and an "imbalance" in the ratio of RP transcripts to other mRNA species. They perform targeted and unbiased profiling of transcripts associated with the polysome, and these experiments are perhaps the major strength of the paper. RPL2 mRNA is excluded from the polysome in the WT, and it is retained in the *ccr4* mutant. Deep sequencing of the polysome-associated mRNA species was performed for the WT and *ccr4* mutant at different temperatures. Their data suggest that RP transcripts do not prevent new transcripts from entering the polysome. The results do suggest that there is a global reprogramming of polysome-associated transcripts in response to temperature stress, and this reprogramming is prevented in the absence of Ccr4. It is not a truly novel finding, but rather a careful characterization of this process in this species. This data likely mirrors prior data sets indicating total cellular transcriptional profiles of these strains incubated at different temperatures.

They also demonstrate that antigen masking is defective in the *ccr4* mutant at elevated temperature. These studies confirm prior work demonstrating the dynamic nature of the *C. neoformans* cell surface in response to host-relevant conditions. This results in the unmasking of the immunogenic beta-1,3-glucan component of the cell wall in a subpopulation of the *ccr4* mutant, but not in the WT. This particular strain was also more readily phagocytosed *in vivo*, perhaps as a result of the altered immunogenicity of its exposed cell wall. Similar cell wall changes

can be seen as a compensatory response to many cell stresses. Their data suggest that temperature stress can induce these changes, and that Ccr4-mediated mRNA decay helps the fungal cell in its active remodeling of the cell wall in an adaptive manner. They have done a very nice job in showing how a biologically important phenomenon, stress-mediated alterations in ribosome occupancy, control host-pathogen interactions in a pathogenic microbe.

Minor points

The protein, and not the gene, designation for Ccr4 should be used in the 4th line of the 2nd paragraph of the Intro.

They note a reasonable but not full amino acid conservation between Ccr4 in the two *Cryptococcus* species. Would an allele swap between the CCR4 locus of *C. amyloletus* and *C. neoformans* result in full function?

They mention MBL, but the context is not completely clear, not adequately discussed. Similarly, the experiments using sorbitol as an osmo-remediative agent are a bit loose and need a more focused interpretation and discussion.

We thank the reviewers for their insightful comments and suggestions, and feel that the manuscript is greatly improved, tells a more cohesive and complete story, and provides a mechanistic understanding of the connection between mRNA decay and antigen masking.

The initial decision letter suggested that additional mechanistic insight was required. We have included the following new mechanistic observations that we feel increase the impact of our manuscript:

1. We demonstrate that the Ccr4 enzyme from *C. amyloletus* is able to complement the temperature sensitivity of the *C. neoformans ccr4D* mutant. This tells us that the temperature defect in *C. amyloletus* is not due to lack of Ccr4 activity, but is rather a result of a non-responsiveness to temperature stress, leading to the absence of translome reprogramming.
2. We demonstrate that in the absence of mRNA decay, there is a defect in transcriptome reprogramming that is likely due to failure to translate transcription factors required for stress adaptation.
3. Screening mutants of transcription factors repressed in *ccr4D*, we identified *HOB1* as a regulator of unmasking, and demonstrate that *HOB1* is indeed down-regulated in the *ccr4D* mutant during temperature adaptation, linking a dysregulation of *HOB1* to the unmasking phenotype of *ccr4D*.

Reviewer #1 comments:

1.) If the RP transcript decay properly occurs in *C. amyloletus*, will this non-pathogenic fungus become as tolerant to high temperature and pathogenic as *C. neoformans*?

This is a great question, and one that we ask ourselves often. We do not claim that RP repression is the sole driver of host-temperature adaptation, but we do show that it is a response that is lacking in the non-pathogen and thus, the response seems to be *correlated* to adaptation. We feel strongly that experiments that could potentially make *C. amyloletus* adapt to increasing temperatures to study this response lead to unnecessary risk, however minimal, of conferring human pathogenic potential to a non-pathogen. This would be a breach of our ethics.

2.) It is not clear whether the RP transcript decay difference between *C. neoformans* and *C. amyloletus* results from different functions of Ccr4 orthologs per se or other regulators upstream of Ccr4. The former hypothesis could be proven by allele swapping experiment. The latter hypothesis could be proven only after the detailed mechanistic details of the Ccr4-mediated signaling pathway is elucidated.

We thank the reviewer for these suggestions. To address this concern we have performed the allele swap by placing the *C. amyloletus CCR4* gene into the *C. neoformans ccr4Δ* mutant. In Figure 2 of the revised manuscript we have shown that the *C. amyloletus* enzyme is able to restore thermotolerance as well as RP repression in response to host temperature, although not to the levels of WT *C. neoformans*. Our data suggests that there are upstream regulators of mRNA decay that are likely absent, not activated, or have evolved different functions in *C. amyloletus* in response to host-temperature. We are currently working on analyzing some candidate upstream regulators of mRNA decay and plan to assess the response to host-temperature in *C. amyloletus* on a global scale. At this time, we feel that this work is outside of

the scope of the current manuscript.

3.) In Figure 1 and other figures containing northern blot data, the authors need to quantitatively measure gene expression levels by using different loading controls. rRNAs are not appropriate for quantitative measure as they are too abundant (about 80% of total RNA) and their signals are too strong to be used as quantitative loading controls. Instead actin, tubulin or other housekeeping genes could be used. Otherwise, they should provide quantitative RT-PCR data with three biological independent replicates and perform statistical analysis to determine whether any difference is meaningful or not. Currently, the authors did not perform any statistical analysis. Based on the Figure 1a northern blot, there seems to be a difference in transcript levels between zero time and 3 hr in *C. amyloletus*. In addition, the label “WT” should be written as “*C. neoformans*”, because *C. amyloletus* is also WT in the given species.

We appreciate the reviewer’s concern on this subject. Standard housekeeping genes such as actin and tubulin do not remain unchanged in response to host-temperature stress. Like RP transcripts, which are often considered to be “housekeeping”, they are repressed, and therefore are not suitable controls. In addition, deadenylation dependent decay has the potential to regulate ANY pol II transcript to some extent, so by selecting a non-polyadenylated RNA as a loading control, we bypass any potential unforeseen effects of deadenylation on control mRNAs. By utilizing rRNA, we are essentially measuring the quantity of RP transcripts over time in comparison to total RNA. Further, rRNA normalization is largely standard in the mRNA decay and polysome profiling fields, and has been used by several papers published in Nature Communications in the last year:

Blasco-Moreno *et al.* The exonuclease Xrn1 activates transcription and translation of mRNAs encoding membrane proteins. *Nature Communications* **10**, Article number 1298 (2019)

Kim *et al.* Intracellular interleukin-32 γ mediates antiviral activity of cytokines against hepatitis B virus. *Nature Communications* **9**, Article number 3284 (2018);

Ahn *et al.* nc886 is induced by TGF- β and suppresses the microRNA pathway in ovarian cancer. *Nature Communications* **9**, Article number 1166 (2018)

We have performed ANOVA analysis to compare levels of *RPL2* at each time point to t=0 to identify significant changes in its expression. In regards to Figure 1a, our other replicates do not indicate a drop in *RPL2* at t=3 hours. We have exchanged the images with a different replicate that better represents the data.

In all instances where the two wild type species, *C. neoformans* and *C. amyloletus*, are being compared we have properly labeled them as such.

4.) Figure 2a is hard to understand. What do these graphs mean? There is no definition for Y- and X-axes. There is no labeling on each lane of northern blot data. Based on the rRNA loading controls, they are not equally loaded. In fact, Figure 2a was described later than Figure 2b in the main text. Also Figure 2C is described later than Figure 3a. So in general the order of figures should be rearranged according to the description order in the main text.

We apologize for the confusion brought on by this figure. We have properly labeled the axes on Figure 2a, which shows polysome traces for each of our strains obtained by measuring the Absorbance at 254 nm of lysates that were separated in a sucrose gradient. We have also indicated the subunits and the polysomes in the trace for clarification. The northern blot and rRNA images below the traces correspond to the fractions that were simultaneously collected and we have adjusted the Figure legend to make this clear. The northern analysis is a qualitative assessment of the distribution of the *RPL2* transcript only, and is not normalized to the rRNA. The rRNA images are included to show the presence of the small subunit, large subunit, and ribosomes throughout the gradient.

We have reorganized our figures so that they are discussed in order.

6.) The quality of Figure 4a, 4c and 5a is too low. The colored fluorescence images could be replaced with better ones. For example, the result of Figure 5a does not correspond to the Figure 5b. In the Figure 5a, the fluorescence signals bound by Dectin-1 in *C. amyloletus* at 30C appear to be identical to those at 37C, which does not correspond to the result of Figure 5b.

Due to the size of these figures the images were scaled down. We have created magnified insets to better display our results. In regards to Dectin-1 in *C. amyloletus* in Figure 5a, the fluorescence at 30°C is non-specific (similar to our no-primary controls) and never showed cell wall specific staining at the periphery like we see at 37°C. We hope that this is clarified by the insets that we have provided.

7.) The graph of Figure 6c requires statistical analysis data to indicate which samples are significantly different.

Due to the changes that we have made in the manuscript, including the allele swap and the transcription factor investigation, we felt that the cell cycle section was not cohesive with the rest of the revised manuscript and have removed it.

8.) Page 12, line 1: Figure 5b should be Figure 5c.

This has been edited.

9.) On page 12, the authors mentioned that the *ccr4* mutant shows cell cycle defects following host temperature stress. Based on the figure 6a, the *ccr4* mutant is already defective in cell cycle regulation even under unstressed conditions (30C). So the role of Ccr4 in cell cycle regulation is not a high-temperature specific phenotypic trait.

We appreciate the reviewer's observation, and we agree that cell cycle is defective in the absence of Ccr4 even when no stress is present. The addition of stress does appear to make this defect worse. As mentioned above, we have removed the cell cycle section from this manuscript.

Reviewer #2 Comments:

10.) The protein, and not the gene, designation for Ccr4 should be use in the 4 line of the 2nd paragraph of the Intro.

This has been edited.

11.) They note a reasonable but not full amino acid conservation between Ccr4 in the two *Cryptococcus* species. Would an allele swap between the CCR4 locus of *C. amyloletus* and *C. neoformans* result in full function?

We thank the reviewer for this suggestion. We have performed the allele swap experiments (Figure 2) and have concluded that the *C. amyloletus* Ccr4 is able to restore thermotolerance and RP repression in the *C. neoformans ccr4Δ* mutant.

12.) They mention MBL, but the context is not completely clear, not adequately discussed.

We have trimmed this section of the discussion for word limitation, and it is no longer in the manuscript.

13.) Similarly, the experiments using sorbitol as an osmo-remediative agent are a bit loose and need a more focused interpretation and discussion.

As mentioned above, due to the changes that we have made in the manuscript, including the allele swap and the transcription factor investigation, we felt that the cell cycle section was not cohesive with the rest of the revised manuscript and have removed it.

Reviewers' comments:

Reviewer #1 (Remarks to the Author):

This reviewer raised several major and minor concerns for the previous version of manuscript. In particular, I criticized a lack of any mechanistic insight into how Ccr4-related signaling pathway mediates host-temperature adaptation, cell wall remodeling, and cell cycle regulation, mainly because this is a continuing story of the Panepinto group about Ccr4 and the pleiotropic roles of Ccr4-Not transcriptional complex in *S. cerevisiae*. In response to my comments, the authors provided two additional data in this revised manuscript. First, heterologous expression of *Cryptococcus amyloletus* CCR4 partially restored temperature sensitivity of the *Cryptococcus neoformans* ccr4 mutant. Second, using the transcription factor mutant library previously constructed by the Bahn group, the authors found that Hob1 is one of downstream transcription factors for Ccr4 and indeed involved in high temperature-mediated exposure of cell wall glucan. Based on these data, which are nice additions (I appreciate it), they claimed that other hitherto unknown signaling regulators upstream of Ccr4 involved in RNA decay activation upon temperature upshift are absent in *C. amyloletus*. However, another interpretation is possible. It is also possible that the Hob1-like transcription factor could be missing in *C. amyloletus*. Therefore, either upstream or downstream factors of Ccr4 could be missing or properly activated (regulated) upon temperature upshift. Therefore, it seems to me that the regulatory mechanism of Ccr4-related signaling cascades is still missing (although it was much more improved than before).

The authors addressed most of minor comments that this reviewer made. However, the authors need to clarify that *C. amyloletus* CCR4 was integrated into the native locus of *C. neoformans* CCR4 or ectopically integrated. Although they mentioned that they performed the northern blot and southern blot analyses, I can't find the data. Furthermore, it was rather disappointing that none of complement strain data is included in this paper. Particularly, the CCR4 allele swapping experiment should have been done with the proper ccr4: :CCR4 complemented strain as a control.

Reviewer #2 (Remarks to the Author):

The authors have submitted a revised manuscript regarding transcript stability of ribosomal protein genes as a mediator of thermotolerance in the human fungal pathogen *Cryptococcus neoformans*. They have responded to prior reviews with new experiments, including allele swap of the CCR4 gene between the pathogenic *C. neoformans* and the related non-pathogen *C. amyloletus*. They also provide evidence that the transcriptional regulator Hob1 is involved in regulating this rapid cellular response to thermal stress.

The authors have addressed all the issues raised in the prior review. This work will certainly direct new work in the mechanisms of pathogen adaptation to the host environment.

Reviewer #1 comments:

...(the authors) claimed that other hitherto unknown signaling regulators upstream of Ccr4 involved in RNA decay activation upon temperature upshift are absent in *C. amyloletus*. However, another interpretation is possible. It is also possible that the Hob1-like transcription factor could be missing in *C. amyloletus*. Therefore, either upstream or downstream factors of Ccr4 could be missing or properly activated (regulated) upon temperature upshift. Therefore, it seems to me that the regulatory mechanism of Ccr4-related signaling cascades is still missing (although it was much more improved than before).

We thank the reviewer for this interpretation and agree that it is possible and highly likely that additional downstream signaling events maybe missing or not activated upon host temperature stress in the non-pathogen. At this point we are not sure if RP repression precedes or is necessary for the cell wall remodeling to occur or if it is a separate, unrelated Ccr4-mediated event. These are all things we hope to understand in our future endeavors. We have revealed these uncertainties in the discussion and have mentioned that downstream components also need to be investigated.

The authors need to clarify that *C. amyloletus* CCR4 was integrated into the native locus of *C. neoformans* CCR4 or ectopically integrated. Although they mentioned that they performed the northern blot and southern blot analyses, I can't find the data.

We have revised the methods section to specifically declare that the construct was randomly integrated into the genome of the mutant strain. In addition we have provided northern and Southern blot images in Supplemental Figure 2. Due to the inefficiency of the *C. amyloletus* CCR4 probe being able to detect *C. neoformans* CCR4 and vice/versa we provided the northern blot probed with both probes in Figure 2 to show expression of CCR4 in all strains tested. Both probes are also included for the southern blot in the uncropped images in the source data for Figure S2.

Furthermore, it was rather disappointing that none of complement strain data is included in this paper. Particularly, the CCR4 allele swapping experiment should have been done with the proper *ccr4::CCR4* complemented strain as a control.

We thank the reviewer for raising this concern, and agree that comparison between the two complements was a nice addition. We have completely revised Figure 2 to include data from the *C. neoformans* complement strain as well. You will see that while the *C. neoformans* CCR4 complements growth RP and destabilization comparable to wild type, the *C. amyloletus* CCR4 demonstrates intermediate levels of complementation of these phenotypes.